



# High resolution observations of small scale gravity waves and turbulence features in the OH airglow layer

René Sedlak[1], Patrick Hannawald[2], Carsten Schmidt[1], Sabine Wüst[1], Michael Bittner[1,2]

[1]German Aerospace Centre, Germany – German Remote Sensing Data Centre

[2]University of Augsburg, Germany – Institute of Physics

*Correspondence to*: René Sedlak (rene.sedlak@dlr.de)

**Abstract.** A new version of the Fast Airglow Imager (FAIM) for the detection of atmospheric waves in the OH airglow layer has been set up at the German Remote Sensing Data Center (DFD) of the German Aerospace Center (DLR) at Oberpfaffenhofen (48.09° N, 11.28° E), Germany. The spatial resolution of the instrument is 17 m in zenith direction with a

field of view (FOV) of 11.1 km x 9.0 km at the OH layer height of ca. 87 km. Since November 2015, the system has been in operation in two different setups (zenith angles 46° and 0°) with a temporal resolution of 2.5 to 2.8 s.

In a first case study we present observations of two small wave-like features that might be attributed to gravity wave instabilities. In order to spectrally analyse harmonic structures even on small spatial scales down to 550 m horizontal wavelength, we made use of the Maximum Entropy Method (MEM) since this method exhibits an excellent wavelength

resolution. MEM further allows analysing relatively short data series, which considerably helps to reduce problems such as stationarity of the underlying data series from a statistical point of view. We present an observation of the subsequent decay of well-organized wave fronts into eddies, which we tentatively interpret in terms of an indication for the onset of turbulence.

Another remarkable event which demonstrates the technical capabilities of the instrument was observed during the night of

4th to 5th April 2016. It reveals the disintegration of a rather homogenous brightness variation into several filaments moving in different directions and with different speeds. It resembles the formation of a vortex with a horizontal axis of rotation





likely related to a vertical wind shear. This case shows a notable similarity to what is expected from theoretical modelling of Kelvin-Helmholtz instabilities (KHIs).

The comparatively high spatial resolution of the presented new version of the FAIM airglow imager provides new insights

into the structure of atmospheric wave instability and turbulent processes. Infrared imaging of wave dynamics on the sub-kilometre scale in the airglow layer supports the findings of theoretical simulations and modellings.

# 1 Introduction

The reaction of hydrogen and ozone produces molecular oxygen and vibrational-rotationally excited hydroxyl (OH)*. OH*

relaxes by emitting electromagnetic radiation, especially in the SWIR range (Bates and Nicolet, 1950). The hydroxyl forms a

Chapman layer at the upper mesosphere / lower thermosphere and is one prominent component of the airglow phenomenon.

The peak concentration of OH* is located at an altitude of ca. 87 km with a full half-width of about 8 km (Baker and Stair,

1988). The OH* mean emission altitude exhibits low annual variability (e.g. von Savigny, 2015). Since the density of the

atmosphere decreases exponentially with increasing height, the amplitudes of upward propagating atmospheric waves

considerably grow due to the conservation of energy. Once the waves reach the airglow layer, they influence the intensity of

the airglow emission. This makes the airglow an established phenomenon for the investigation of atmospheric dynamics.

Furthermore, based on temperature data derived from a Na-lidar, (Gardner et al., 2002) concluded that the vertical heat flux

due to wave dissipation is maximum near the mesopause altitude region because of the reduced stability due to the large

temperature lapse rate and the low buoyancy parameter below the mesopause. It is thus expected to frequently observe

gravity wave breaking processes in this altitude region.

Gravity waves are an intensively investigated feature within atmospheric dynamics. Their ability to transport energy and

momentum over long distances through the atmosphere makes them an important factor to be considered in climate studies

and meteorology. It is well known that gravity wave spectra extend over a few magnitudes of the wavelength scale.

Horizontal wavelengths from a few kilometres up to about thousand kilometres are usually observed.

Energy transported by atmospheric waves is released into the surrounding atmosphere when instabilities lead to turbulence.

Atmospheric instability can either be of dynamical or of convective nature. Dynamical instabilities are caused by wind shear,





leading to formation of Kelvin-Helmholtz instabilities (KHI), whereas convective instabilities are induced when the temperature lapse rate gets superadiabatic (Fritts and Alexander, 2003).

Turbulent processes are characterized to cover several magnitudes of the length scale. The complete range of turbulence can be divided into three parts: within the largest regime, the so-called buoyancy subrange, energy is still carried by waves and

the buoyancy force is dominating viscosity. Distortions on scales below the buoyancy subrange let the transported energy cascade to smaller and even smaller structures (inertial subrange) where energy is still conserved within the eddies until it is dissipated into the atmosphere due to viscous damping (viscous subrange) (e.g. Lübken et al., 1987). It must be noted that the scale limits between the three subranges are strongly dependent on the altitude. Within the abovementioned region of the OH* peak concentration at an altitude of about 87 km, the inertial subrange of turbulence is on a scale of a few hundreds of

metres, starting just below 1 km and reaches the viscous subrange at about 30 m (Hocking, 1985).

Signatures of atmospheric gravity waves have already been extracted from time series of OH rotational temperatures which can be derived from airglow emission spectra (e.g. Scheer, 1987; Takahashi et al., 1999; Schmidt et al., 2013; Wüst et al., 2016).

In order to deduce horizontal information of atmospheric waves, infrared camera systems have come up as a highly

promising remote sensing technique (see e.g. Peterson and Kieffaber, 1973; Herse et al., 1989; Moreels et al., 2008) as they allow obtaining images of atmospheric waves. Gravity wave activity at airglow altitude of the entire night sky can be imaged using all sky lenses (Taylor et al., 1997; Smith et al., 2009 and many others). Another application for airglow imagers is the investigation of small scale gravity wave structures, such as acoustic-gravity waves (e.g. Nakamura et al., 1999) or instability features (Hecht et al., 2014). A high spatio-temporal resolution is required for this purpose, which is achieved at the expense

of the aperture angles of the imager. However, theoretical calculations especially in the spatial regime below 1 km are hardly supported by experimental observations yet, since a better spatial resolution than 500 m is needed to image structures on these scales (Hecht et al., 2014).

As it has been reported by (Hannawald et al., 2016), the infrared imaging system FAIM (Fast Airglow Imager) is currently being operated by the German Aerospace Center (DLR) in order to continuously observe gravity waves in the OH airglow

each night with a temporal resolution of 0.5 s. Obtaining a mean spatial resolution of 200 m, small scale gravity wave



structures with a wavelength down to 2 km have already been discovered. Such instruments are currently operated at the Sonnblick Observatory (47.05° N, 12.97° E) in Austria (FAIM 1) as well as at Oberpfaffenhofen (48.09° N, 11.28° E), Germany (FAIM 4) for routine observations within the international Network for the Detection of Mesospheric Chance (NDMC, http://wdc.dlr.de/ndmc). A third system of this kind, FAIM 2, is installed for air-craft based observations on the

DLR research aircraft FALCON.

In order to investigate even smaller scale gravity wave signatures and turbulent features within the inertial subrange at mesopause heights a further developed FAIM system, FAIM 3, with a considerably improved spatial resolution is presented in this paper.

## 2 Instrumentation

FAIM 3 is the second version of the already mentioned Fast Airglow Imagers established at DLR (Hannawald et al., 2016). It is based on the SWIR camera CHEETAH CL developed by Xenics nv. Its 640 x 512 pixels InGaAs sensor array with a pixel size of 20 µm x 20 µm is sensitive to the spectral range from 0.9 to 1.7 µm. The raw signal is converted into a 14 bit digital stream, which is sent to a frame grabber via CameraLink. The sensor is thermoelectrically cooled down to an operational temperature of 233 K. The dissipated heat is carried away by a closed circuit water cooling system. A schematic

drawing of the entire setup is shown in Figure 1.

All measurements presented here are performed at DLR at Oberpfaffenhofen (48.09° N, 11.28° E), Southern Germany. A 100 mm SWIR lens by Edmund Optics® is used. Its aperture angles have been determined to be 7.3° in horizontal direction and 5.9° in vertical direction. In the first measurement configuration the camera was mounted on a fixed stage with a zenith angle of 46°. Simple geometric considerations reveal that, using the above mentioned lens, the observed area corresponds to

a trapezium-shaped region of the airglow layer, with the height of the trapezium being 18.6 km and a width of 15.2 km to 16.9 km (see (Hannawald et al., 2016) for details of the calculation). We note that the curvature of the Earth has been neglected within these considerations. Thus, horizontally an area of about 299 km² is observed with a mean spatial resolution of 30 m. The camera is adjusted to an azimuth angle of 214°. In order to record a sufficiently strong airglow signal the integration time of the camera is set to 2.5 s. The instrument in this configuration has been in operation from 18th to 30th



November 2015, corresponding to 12 measured nights. The data of the first shown case study was measured with this configuration.

In a second measurement configuration the FAIM 3 zenith angle is adjusted to 0°. Still using the 100 mm lens, a 11.1 km x 9.0 km rectangular area of the airglow layer is acquired. This leads to a mean spatial resolution of 17 m. The integration time has to be raised to 2.8 s since the airglow intensity decreases with decreasing zenith angle due to the van-Rhijn effect (van

Rhijn, 1921).

A flat field correction is applied in order to remove the fixed pattern noise and the vignetting of the acquired images.

As outlined above the section of the sky observed by the FAIM 3 instrument in zenith position is limited to an area of about 100 km². In order to better judge on the overall dynamical situation we operated the FAIM 3 zenith measurements simultaneously with the FAIM 4 instrument where FAIM 4 was allowed to sense the overall sky. The temporal resolution of

FAIM 4 in this case was 1.5 s while the horizontal resolution is roughly 400 m within the image centre.

**3 Data Analysis**

The horizontal wavelengths of gravity waves in an image can for example be estimated by analysing the data series of the pixel intensities along a cross section perpendicular to the wave fronts. The distortion resulting from the abovementioned trapezium-shaped FOV projected onto rectangular images has to be corrected prior to analysis, so that an equidistant metric

scale is provided. The images are mirrored and rotated into the proper azimuthal direction, so that they can be rendered true to scale onto a map (see Figure 2). This correction is performed using the same algorithm as described in (Hannawald et al., 2016). As a typical example, Figure 2 a - f shows a sequence of such images taken on 18th November 2015, between 22:59:55 and 23:04:11 UTC.

In the image a wavelike structure reveals as more or less periodic brightness variations. This brightness variation clearly

appears in the series of pixel intensities along a direction perpendicular to the wave fronts in the image, as it is shown in Figure 3. Spectral analysis of the data series allows deducing the horizontal wavelengths. Since in our case the observed gravity waves are localized within a comparatively small area of the FOV and exhibit wavelengths of relatively small scales, the data series of pixel intensities extend over less than a hundred data points. Considering also the pronounced noise level,

the Maximum Entropy Method (MEM) turned out to be a powerful technique to estimate the spectral density of the data sets

(Ulrych and Bishop, 1975), even if the data series is so short. As an example, Figure 4 shows the MEM spectrum of the data

series shown in Figure 3.

The MEM provides an approach for estimating the spectral density of a time series with a high frequency resolution (e.g.

Bittner et al., 1994). While other methods like the wavelet analysis try to fit a predefined function to the data, the MEM does

not claim any assumptions about the theoretical shape of the signal but to be a stationary Gaussian process. According to

(Jaynes, 1963) the most probable of all possible spectra describing a dataset is the one which makes the least assumptions

about the information the data contain. This corresponds to the state of maximum entropy. The MEM derives the most

probable spectrum of a time series by fitting an autoregressive model (AR) of order N to the data. The AR coefficients are

determined using Burg's algorithm, which calculates them based on forward and backward predictions of the data. The

major issue of the MEM is to find the right order N of the underlying AR process. If the order is chosen too small several

frequency peaks may not be resolved, whereas an overestimated order will lead to further splitting of the peaks into spurious

subpeaks due to noise approximation (e.g. Wüst and Bittner, 2006).

## 4 Results

During the night from 18th to 19th November 2015 observations were performed at a zenith angle of $46°$, leading to a

spatial resolution of approximately $30\,m$. In order to demonstrate the performance of the instrument, one interesting gravity

wave event, registered between 22:59:25 and 23:04:33 UTC, is presented. The relevant period is shown in parts by a series

of six images in Figure 2a–f and in its entirety in Video 1 in the supplemental material. First order difference images (that

means the temporal derivation of the original signal, here $\Delta t = 7.5\,s$) are shown here to help better visualize wave structures.

The difference of two images acts as a high pass filter, suppressing long-period oscillations and highlighting the regions with

varying intensity between the two images. However, the original data is used, of course, for spectral analysis in order to

retrieve the correct spatial content. It is interesting to note that obviously a smaller scale wave packet is propagating from the

left corner of the FOV (West) to the right corner (East). The packet, which is marked by white circles in Figures 2a–e,

moves in the same direction as a second, larger wave structure but its wave fronts are tilted against each other horizontally



by an angle of about 45°. The larger wave structure extends nearly over the entire image from the lower left to the upper

right corner of Figures 2a–f and the quite limited horizontal width of the wave fronts makes the wave appear to be trapped

in a narrow corridor in the airglow layer. Having nearly completely passed through the FOV the wave packet seems to

decompose into some disordered features as time proceeds and is no longer visible in Figure 2f. This is best recognizable at

the end of Video 1, starting at 23:03:20. Figure 3 shows in black the pixel intensities along a cross section perpendicular to

the wave fronts of the small wave packet in the image acquired at 23:00:42 UTC (the position of the cross section is given in

Figure 2b). Visual inspection of the image series and the cross section shows that the wavelength of the wave packet is about

half a kilometre. The wavelength of the larger wave structure can be visually estimated as about 1.7 km. For a more detailed

analysis of the smaller structure the MEM is applied to the data series of Figure 3. An order of 20 is chosen as this

corresponds to one third of the number of data points, which has turned out to be a good choice for analysing short data

series with less than 100 values (Bittner et al., 1994). The MEM spectrum in Figure 4 exhibits a strong peak at about 550 m.

It is clearly above the 99 % level of statistical significance and fits to the small scale wave packet discovered in the images as

already mentioned above. It is interesting to note that the MEM spectrum also reveals four other significant peaks at

wavelengths well below 200 m, which can also be found in the signal of Figure 3. In the following we will focus on the

550 m wave structure, as it is clearly recognizable in Video 1.

It can be seen in the image series that the 550 m wave signal is visible in the examined cross section from 23:00:17 to

23:00:55 UTC. In a next step the same cross section as above is analysed for each image to derive some insight about the

spatio-temporal evolution of the structure. Figure 5 shows the temporal development of the MEM spectrum. The order has

been raised to 36 ensuring that signatures of wavelengths below 1 km are fully resolved. A stable maximum in the

spectrogram (Figure 5) around a horizontal wavelength at about 550 m is obviously present over almost the entire time range.

Another interesting event was observed during the night from 4th to 5th April 2016. The images (see Figures 6a–f) have

been acquired at zenith position, ensuring a spatial resolution of about 17 m. The dimensions of each image are

approximately 11.1 km x 9.0 km. The zenith images show the sky as it is seen from the ground looking upwards. The upper

image side corresponds to an azimuthal direction of 303°. Video 2 shows the entire event. A wave front, indicated by the

dashed red line to guide the eye in Figure 6a, enters the FOV in the upper right corner. The straight red arrow shows the



direction of propagation. A new structure develops at 03:17:48 UTC (Figure 6b), marked by the ellipse in the upper right

corner. At 03:18:27 UTC (Figure 6c) some filament-like structures of different velocities arise in the area of the dashed

orange box. Video 2 shows what is hardly recognizable in the displayed snapshots of Figure 6: the filaments start forming a

vortex structure (emphasized by the curved yellow arrows at 03:19:01 UTC, Figure 6d), which still moves in the direction of

the straight red arrow. At 03:19:49 UTC (Figure 6e) the vortex structure begins to decompose into some disorganized

features while the remaining structure continues following the initial direction (straight red arrows).

To put the observations into a larger spatial context, the FAIM 3 data are compared to simultaneous all sky measurements

taken by the FAIM 4 instrument. Since the two cameras are deployed next to each other, the FOV of FAIM 3 is embedded in

the centre of the FOV of FAIM 4 with a tilt of $19°$. The FAIM 4 measurements are presented in Video 3. A snapshot is given

in Figure 7. Besides the normal image (Figure 7a), the difference image (time difference of $60\,s$) is displayed on the right

side (Figure 7b). The approximate FOV of FAIM 3 is indicated by the green boxes in Video 3 and the white boxes in Figure

7. The all sky images reveal a clear and starry sky with high gravity wave activity, which can be determined on the basis of

the characteristic patchy structures. The remarkable structure observed by FAIM 3 can be found again in Video 3 as a bright

feature within the green box, propagating towards its lower left corner. It can be well recognized in the screenshot of Figure

7.

## 5 Discussion

The first of the two case studies presented here shows a wave packet with a horizontal wavelength of about $550\,m$ moving in

the same direction as a $1.7\,km$ wave with the wave fronts being tilted about $45°$ against the other structure. We therefore

tentatively interpret the wave packet as a ripple structure; see e.g. (Peterson, 1979). Ripples are spatially limited wave

structures with a short life time (Adams et al., 1988) and are caused by atmospheric instabilities (Taylor and Hapgood,

1990). As the $550\,m$ wave packet does not form within the FOV, its lifetime cannot be estimated from our data. However,

the appearance and disappearance of the $1.7\,km$ wave is located within the FOV, which allows estimating its lifetime to

about $16\,min$ if our speculation holds. According to (Taylor and Hapgood, 1990) this feature can also be referred to as a

ripple structure. The fact that the spatial extent of the wave fronts perpendicular to the direction motion is much smaller than



the extent of the wave packet in propagation direction is unusual (ratio approximately 1:6). Obviously, it cannot be described as a plane wave (of infinite extent), it may rather be trapped in a duct or the entire pattern, which appears to be a wave structure, may indeed be an instability feature itself, which is not unlikely given the small dimension with an apparent

wavelength of 1.7 km. It may be consequently speculated that the 1.7 km wave itself is a possible instability feature of a larger scale wave with a horizontal wavelength of several kilometres and decays into the 550 m wave packet as a subordinate instability structure. Such dynamics have already been observed on larger scales in (Hecht et al., 2014). Assuming the same scenario here, the ratio of primary to secondary wavelength is about 3.1. This number agrees quite well with earlier KHI models (Klaassen and Peltier, 1991). As described in section 1, KHIs are instabilities caused by wind shear. This could be a

possible explanation for the tilt of the wave fronts.

At around 23:03:30 UTC the 550 m wave packet starts to collapse into some disordered features, which we tentatively assign to turbulence. Following the considerations of (Hocking, 1985), observations on that scale are already situated in the inertial subrange at airglow altitudes. Similar turbulent-looking features resulting of atmospheric instabilities have been found in the measurements of (Hecht et al., 2014) in the buoyancy regime. Related Direct Numerical Simulations (DNS) (see Fritts et al.

2014) to that data have already predicted secondary instability features below 1 km, which had not been able to observe with an airglow imager so far. However, it must be stressed that these interpretations remain speculative at this stage as the focus of this paper is to demonstrate the performance of the FAIM 3 instrument in order to resolve smaller spatio-temporal scale dynamics in the mesopause altitude region. Alternatively, the periodic structures could simply be remarkably small scale gravity waves with wavelengths of 1.7 km and 550 m as well, without being results of atmospheric instability. As a possible

explanation for the narrow shape of the greater wave horizontal winds could then be considered to forbid wave propagation in the respective areas.

The second interesting event detected by FAIM 3 is a vortex-shaped structure, which develops out of filament-like features within a wave front and continues propagating in the initial direction before decaying into some disordered features. The formation of filament-like structures developing different velocities can possibly be led back to local instabilities of the

atmosphere due to wind shear. The filament structure decays into disorganized features, which resembles the turbulent collapse of the wave packet on 18th November 2015. Other airglow observations exhibit similar-looking breakdown events



of gravity wave fronts, like in (Hecht et al., 2014). The effects of KHI dynamics in the airglow layer have been modelled

using DNSs and large eddy simulations in (Fritts et al. 2014). The vortex structure as well as the filament features turns out

to be a typical manifestation of turbulence due to KHIs. The high gravity wave activity of the overall night sky, revealed by

FAIM 4 all sky measurements at the same time, possibly contributes to atmospheric instability by influencing the lapse rate.

The aforementioned wave front is also visible in the all sky images. A close inspection of Video 3 shows that even with this

system's coarse resolution indications for the separation of parts from the bright crest can be identified, which agrees with

the turbulent looking decomposition of the wave crest found in the FAIM 3 measurements.

## 6 Summary and Conclusions

In order to observe smaller scale gravity wave events and instabilities or turbulence features in the metre regime, the

established airglow imaging system FAIM (Hannawald et al., 2016) has been improved with regard to spatial resolution,

using an InGaAs sensor array with the fourfold number of pixels (327680) and a 100 mm SWIR lens manufactured by

Edmund Optics®. The mean spatial resolution of 200 m at a 45° zenith angle and 120 m at zenith position achieved by the

established FAIM system has been increased to 30 m and 17 m respectively. The observed scales are thus well within the

inertial subrange regime of turbulence at airglow altitudes. Measurements have been taken at Oberpfaffenhofen (48.09° N,

11.28° E), Southern Germany at a zenith angle of 46° and an azimuth angle of 214° as well as in zenith position.

Two case studies are presented. On 18th November 2015 from 22:59:25 to 23:04:33 UTC a wavelike structure with a

wavelength of 1.7 km and a smaller feature with a wavelength of 550 m propagate in the same direction. Their wave fronts

are tilted against each other by an angle of approximately 45°. The 1.7 km wave is estimated having a lifetime of about

16 min, which leads to the presumption that it might be an instability feature of a larger gravity wave with a wavelength of

several kilometres which cannot be seen in our small FOV. Following this hypothesis, the 550 m structure could be the

resulting subsequent instability feature. Another possibility is that the 550 m wave packet could be the primary instability

feature caused by a small 1.7 km gravity wave. Both cases agree with the theory of turbulence: when atmospheric

instabilities lead to the breakdown of waves, their transported energy is cascaded to smaller scale structures in the inertial

subrange. These considerations combined with the ratio of the wavelengths, which fits to values of earlier theoretical studies

like (Klaassen and Peltier, 1991), suggest that the event of 18th November 2015 could be triggered by KHIs.

Zenith measurements on 5th April 2016 from 03:15:58 to 03:27:46 UTC exhibit the breakdown of a wave front into a vortex

structure and the subsequent decay into disorganized features, probably due to turbulence. Characteristic dynamics of

filament-like features indicate that instability could be generated by wind shear. The observations look similar to modellings

of KHI development and the consecutive turbulence dynamics of waves in the airglow layer (Fritts et al., 2014).

Comparisons with parallel measurements of FAIM 4 obtaining all sky images reveal the high gravity wave activity all over

the sky at that time, which might have contributed to increased atmospheric instability.

It has been demonstrated that FAIM 3 is able to image the dynamics of gravity waves on scales significantly below 1 km.

This can provide experimental insight into processes like turbulent breakdown of waves or generation of smaller scale wave

structures due to atmospheric instabilities. FAIM 3 observations support earlier simulations of instability dynamics as

presented in (Fritts et al., 2014). Apart from that, FAIM 3 is an ideal addition to all sky imagers, since it provides the

opportunity to closer investigate specific airglow structures with a much higher spatial resolution, as it is demonstrated in the

second case study in this paper.

Operational zenith measurements with a mean spatial resolution of 17 m and a temporal resolution of 2.8 s are being

performed automatically every night since 22th February 2016 at Oberpfaffenhofen, Germany. The data is archived at the

WDC-RSAT (World Data Centre for Remote Sensing of the Atmosphere, http://wdc.dlr.de). FAIM 3 is part of the Network

for the Detection of Mesospheric Change, NDMC (http://wdc.dlr.de/ndmc).

**Acknowledgements**

Parts of this research received funding from the Bavarian State Ministry of the Environment and Consumer Protection by
grant number TUS01UFS-67093.



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





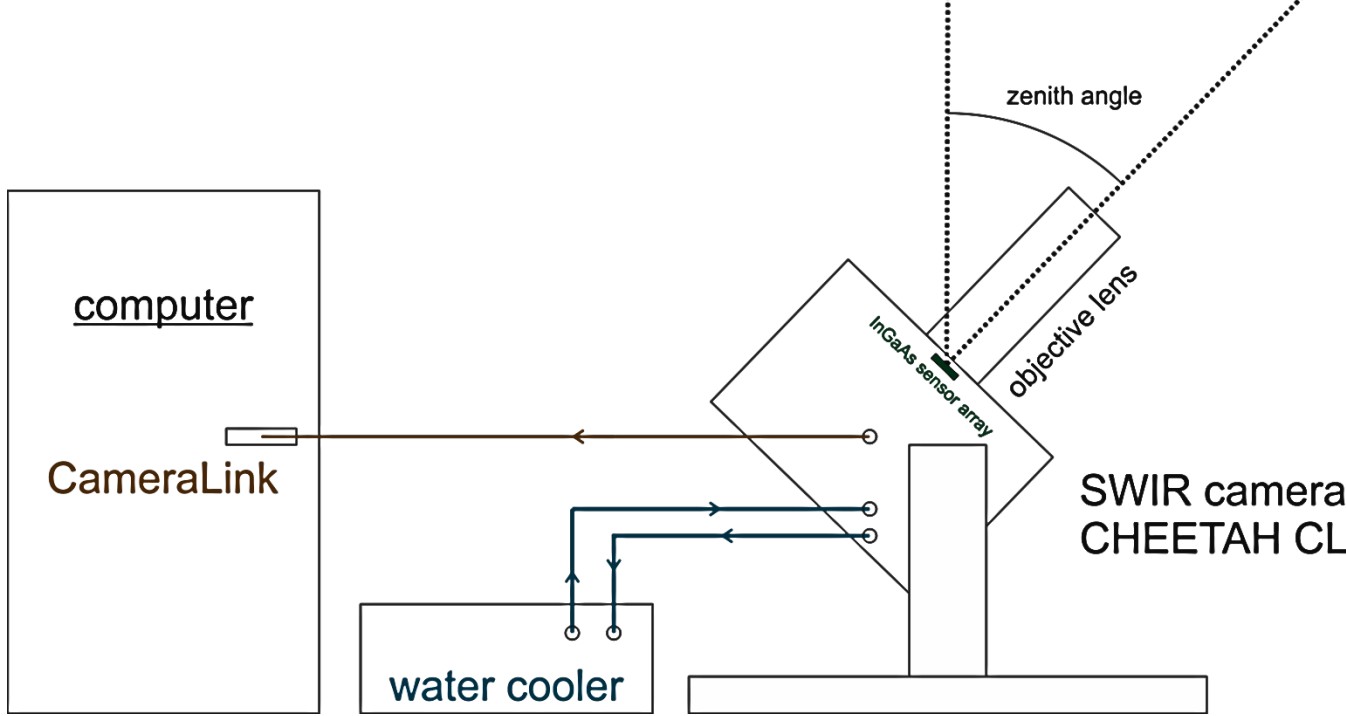

Figure 1. Schematic drawing of the FAIM 3 setup. The measurement configuration with a zenith angle of 46° is shown here.



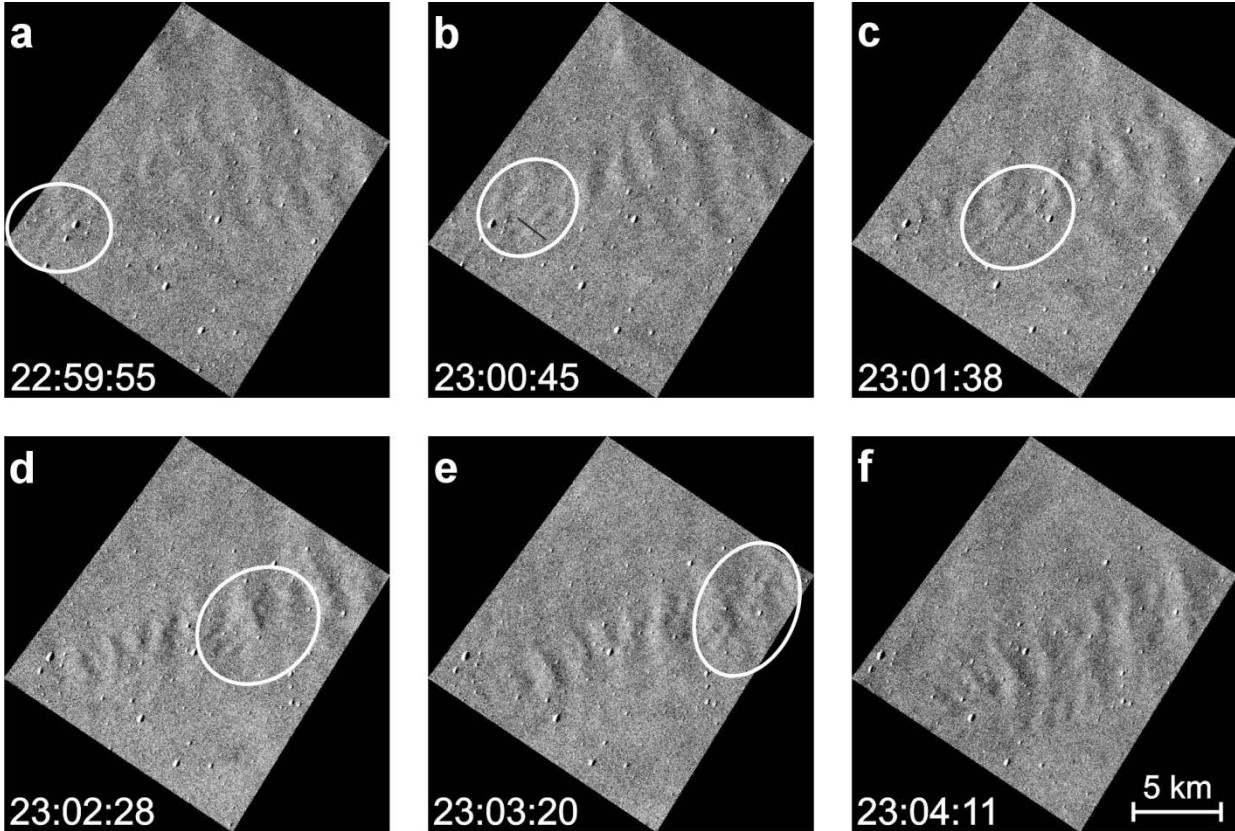

**Figure 2.** Image series of the case study of 18th November 2015 between 22:59:55 and 23:04:11 UTC. Difference images have been chosen for presentation purposes. The images are aligned to the North. The 1.7 km wave structure extends from the western to the eastern corner with several crests being arranged in a narrow corridor. The 550 m wave packet is framed in a white ellipse (images a - e). One may perceive the beginning decay at 23:03:20. The entire sequence is shown in Video 1. The black line, which is shown in image b, indicates the cross section along which the spectral analysis is performed in Figure 3 and 4.





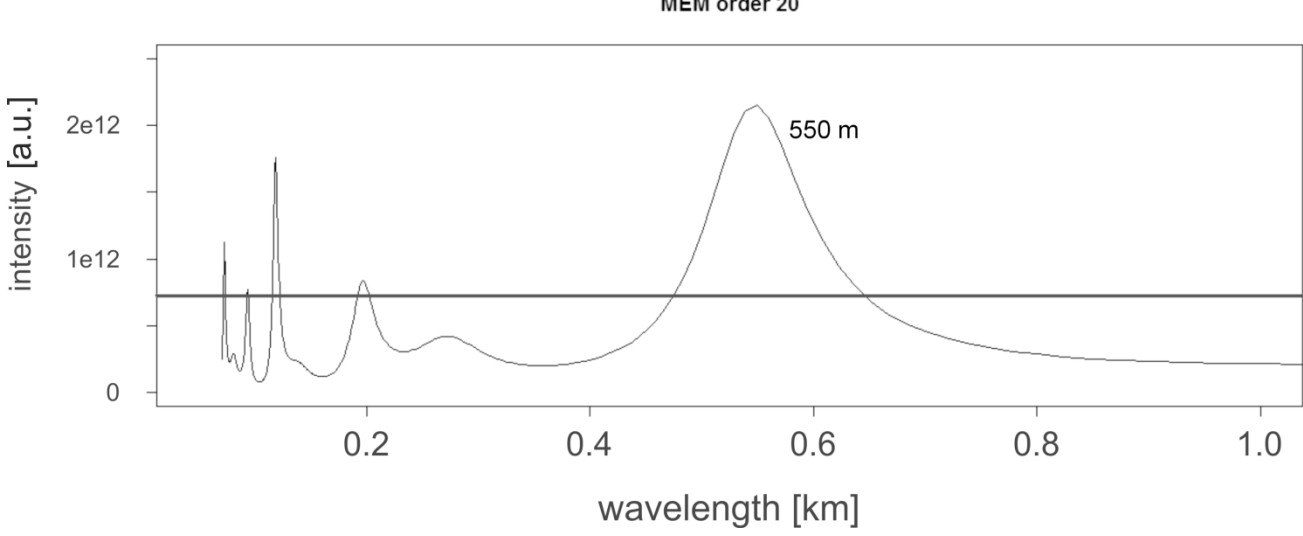

**Figure 3.** Measured intensities along a cross section perpendicular to the wavefronts (its position is indicated by the black line in Figure 2b) in the image of 18th November 2015, 23:00:42 UTC. The pixel range has been converted into kilometres. The orange line shows an ideal 550 m wave (fitted by hand). Only the first half of the data series shows this signal, too.


**Figure 4.** MEM spectrum (order 20) of the data series shown in Figure 3. The horizontal line marks the 99 % level of significance. The peak of the 550 m wave packet is clearly recognizable.





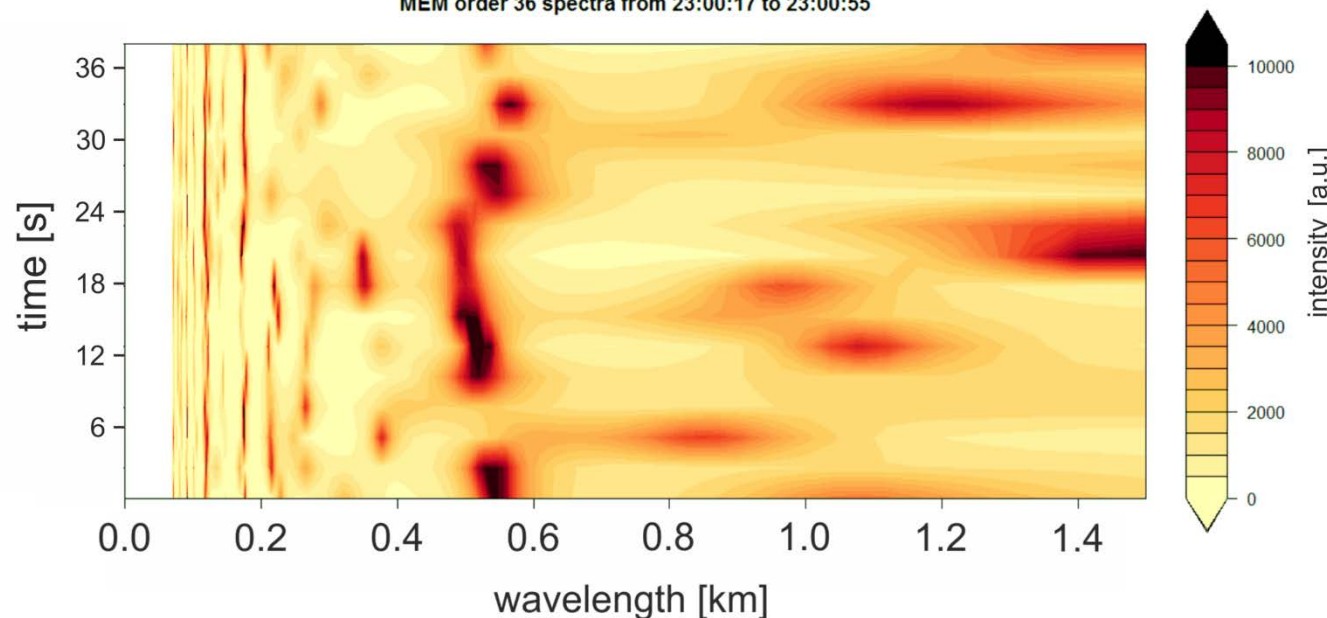

**Figure 5. MEM spectra of order 36 of pixel intensities along the abovementioned cross section (its position is indicated by the black line in Figure 2b) in every image taken between 23:00:17 and 23:00:55 UTC. The signature of the 550 m wave packet is found by the MEM nearly the entire period.**

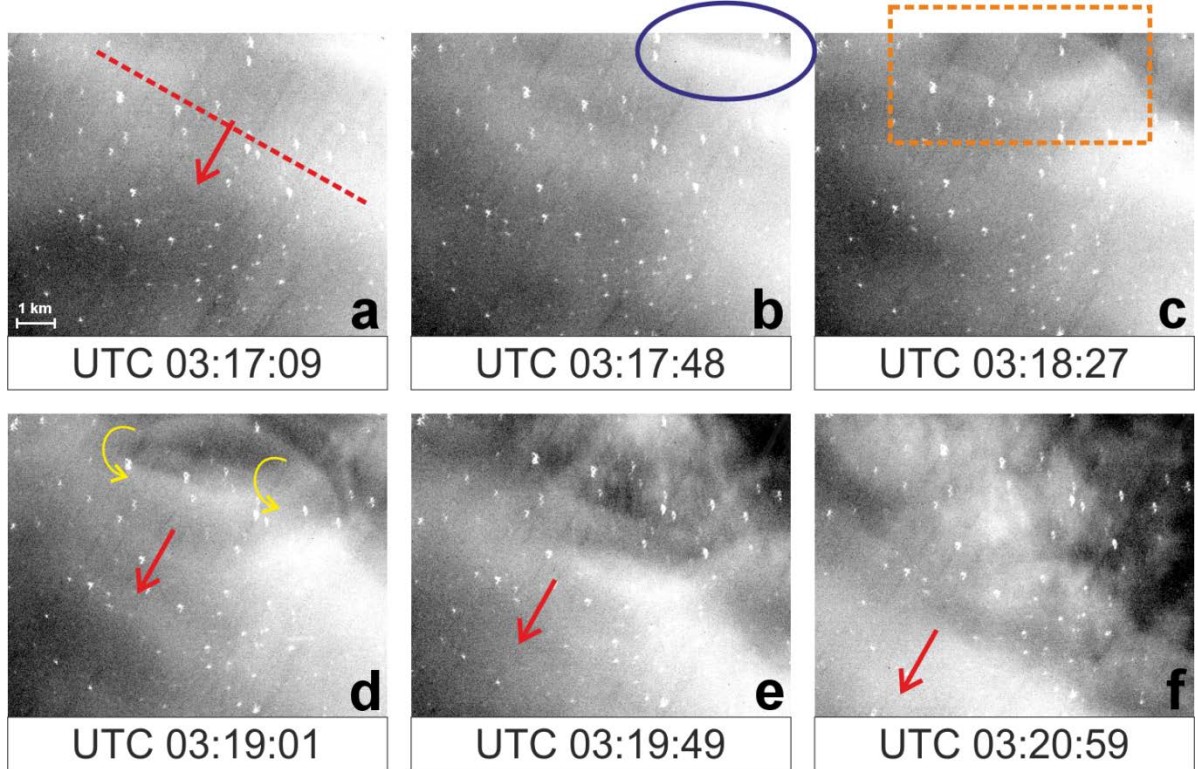

**Figure 6.** Image series of the second case study (5th April 2016 between 03:17:09 and 03:20:59 UTC). A wave front, indicated by the dashed red line, propagates into the direction of the straight red arrow (image a). At 03:17:48 UTC a new structure begins to develop in the elliptically shaped area (image b). Filament-like features arise (dashed orange box in image c) and form a vortex-like structure (curved yellow arrows in image d), which continues propagating into the direction of the initial wave front. The last two images show how the vortex structure decays into disorganized features while the remaining part of the wave front still follows the direction indicated by the straight red arrows (images e and f). The entire sequence is shown in Video 2.





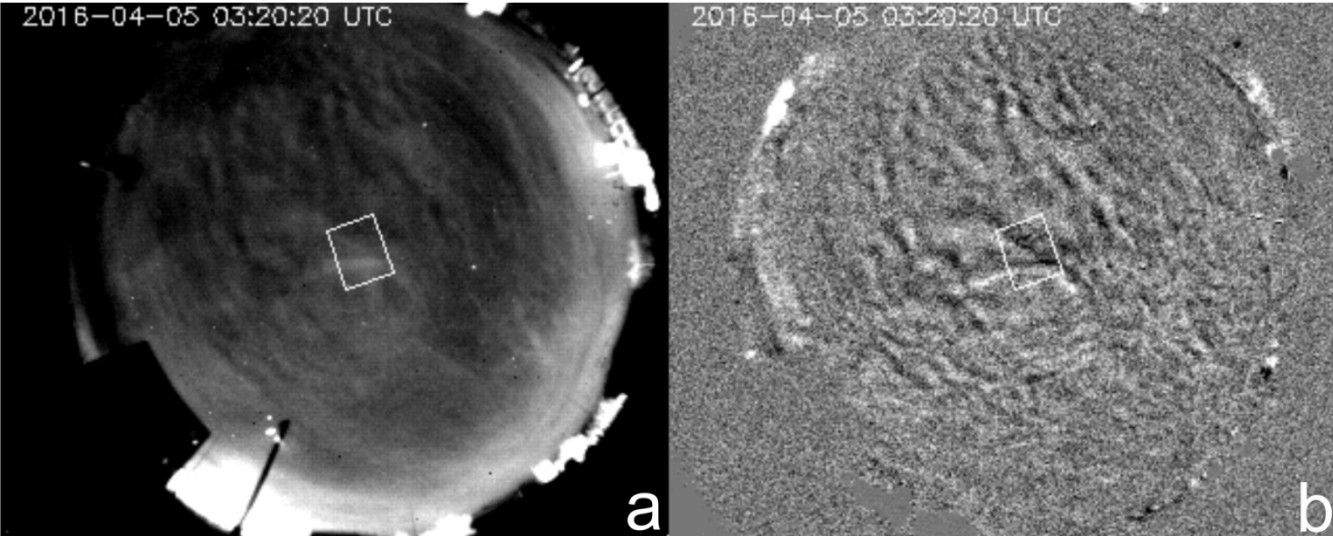

**Figure 7. FAIM 4 all sky image taken on 5th April 2016 at 03:20:20 UTC (left) and respective difference image with a time difference of 60 s (right). The entire sequence is shown in Video 3. The FOV of FAIM 3 is indicated by the white box in the image centres. Due to their spatial structure and their wavelength, we interpret the patchy structures in the starry sky as gravity wave fronts in the airglow layer. The bright feature in the white box probably corresponds to the wave crest that propagates through the FAIM 3 image series at that time (see Video 2 and Figure 6).**
