# Peer review of "High resolution observations of small scale gravity waves and turbulence features in the OH airglow layer"

_Atmospheric Measurement Techniques, 2016_

## Referee Comment (RC1) · Anonymous Referee #1 · 19 Sep 2016

See attached pdf for review.

[Figure]

Review of Sedlak "High resolution observations of small scale gravity waves and turbulence features in the OH layer".

This paper describes a new fast airglow imager with high horizontal resolution. This imager is suitable to study gravity wave related small features. Two case studies are reported. One shows wave fronts broke into eddies. Another shows a homogeneous brightness became filament. The paper is very well written. I would recommend a moderate revision.

The paper could connect better with previous work on "ripples" [Hecht] and turbulence [Yamada] and stress what's new here. It is likely that all the "waves" discussed are not regular gravity waves, but secondary waves already from wave breaking.

When discussing the observations with numerical modeling, more specific discussions are needed.

Minor comments
1. line 29: define SWIR
2. line 39: "frequently observe…" not really from past literature.
3. Line 44: "Energy transported" also momentum.
4. Line 56-58: this paragraph reads odd here. Suggest remove it.
5. Line 92: you mean "299 km²"?
6. Line 140: might the instability (wave) package be advected by the background wind?
7. Line 161-162: "raised to 36" what does this mean?
8. Figure 6: can hardly see the vortex and filament the paper mentioned. Some help is needed.
9. Figure 7: Figure 6 does not fit into the small box in Figure 7. Maybe some rotation is needed?

**Fig. 1.**

---

## Referee Comment (RC2) · Anonymous Referee #2 · 13 Oct 2016

Review of manuscript "High resolution observations of small scale gravity waves and turbulence features in the OH layer" by Sedlak et al.

—————-General Comments—————-

This manuscript reports on a new version of an airglow imager designed for observations of wave features in the hydroxyl airglow layer. The system has been operational since Nov-2015.

Two case studies are presented with analysis of observed wave structures using a maximum entropy method to determine wave parameters in the first case.

If the overall aim of the paper is to demonstrate that the FAIM 3 instrument is able

to image small scale waves (scales <1km) to support numerical modelling studies of (Kelvin Helmholtz Instabilities for example) then this has been achieved.

However, the occurrence of waves in the mesopause region and their breakdown to smaller scale structures (ripples and turbulence) and the deposition of energy and momentum into the mesopause region is well known. I would like to see more discussion on what knowledge gap in these processes can be exploited or investigated with this new instrument development. Elaborate on what is novel (high time and spatial resolution) and how these observations can be used to quantitatively parameterize small scale wave instability and turbulence processes and how these can be used to improve or confirm simulations.

The two events presented both have difficult interpretations. There is a considerable amount of 'tentative' statements, 'possibilities', 'could be', 'might be' and 'speculation' with the interpretation of both events, particularly in the discussion and conclusions. At the end the reader is left with a couple of possibly interesting image sequences of wave events but not really sure what to make of them and what new information they can reveal.

Some careful revision is recommended, a review of the discussion section and expand and elaborate on what new insight into small scale wave dynamics this instrument can reveal and its application in numerical simulations.

—————-Specific comments—————

Line 30. "one prominent component" - The hydroxyl (Meinel) emissions are the brightest of all airglow.

Line 136-137 – I'm not sure the image differencing does help the reader visualize the wave packet in this case study. The 1.7km wave structure is apparent but the 550m wave packet under analysis is rather more difficult to see in Figure 2 (perhaps more apparent in the Video). Does the image have enough dynamic range to contrast stretch

the frames in Fig 2 ?. The use of the word 'obviously' in line 140 is perhaps overstating it.

Line 154 – Despite the level of significance quoted (how is this obtained ? please check), the fit to the pixel count in fig 3 does not look all that significant. At a range of 1.5km for example the fit is completely out of phase with the signal.

Line 155 – I suspect the peaks below 200m are an artefact of the sampling resolution and its harmonics, coupled with detector noise.

Line 168 - 183 The interpretation and description of this second event is speculative not entirely convincing to this reader. In particular the assertion of a 'vortex' structure is very difficult to determine from the 2D image sequence. The interpretation is not aided by the comparison with FAIM 3 as the wavefront propagation does not appear to match. In FAIM 3 (fig 6) the wavefront is parallel to the long axis of the fov. In FAIM 4 (fig 7) the wavefront is propagating top to bottom with wavefront parallel to the short axis of the fov. The wave propagation direction should be given for both instruments to ensure they are consistent. There is not enough resolution in the FAIM 4 sequence to determine any similarity between the two image sequences. Perhaps zooming in or blowing up the FAIM 3 fov in the FAIM 4 image sequence will aid the comparison but the statement from line 180 that the 'remarkable structure' . . . can be found in Video 3 .. and 'can be well recognized in the screenshot' is not valid as presented. I cant !.

—————-Corrections—————-

Line 31 – a 'full half-width' ? – omit full

Line 34 – <-> grow considerably they 'density variations' influence the intensity . . .

Line 36 – omit brackets

Line 46 – reference required for KHI

Line 56 – not sure if this belongs here ? switched from intensity observations to rotational temperature observations ?

Line 71 – a wavelength's'

Line 71 – discovered observed

Line 71 – Such Similar

Line 74 – aircraft

Line 80 – omit 'already mentioned'

Line 84 – 'A schematic diagram of the system' is . . . ?

Line 89 – consider revising .. perhaps "The geometry of this arrangement implies a trapezium-shaped field-of-view at the airglow layer of height 18.6km . . . " ..

Line 92 –An overall area of 299km2 is observed in the hydroxyl layer, not horizontally ?

Line 95 – omit 'corresponding to' "(12 nights)" omit 'shown'

Line 98 – insert (100km2) after rectangular area, then omit line 102

Line 105 – using 'horizontal' resolution here compared to 'spatial' resolution in line 93, 98. Is there a difference ?. Should the units not be 30, or 17 or 400 'm per pixel' (m/pixel) ?

Line 107 omit 'for example'

Line 114 – consider revising this sentence.

Line 115 – How was the direction perpendicular to the wavefront (the black line in fig 2) determined ?.

Line 135 – replace 'relevant period' with 'interval'

Line 164 – at "the" zenith replace 'ensuring' with 'with' 17m/pixel ?

Line 190 – what speculation are you holding ?
Line 209-211 – sentence doesn't make sense ? – revise

Line 218 – omit 'in'

Paragraph from line 221 – as discussed above. I fail to see the similarity as presented, especially with the wave propagation direction issue.

Line 255 – 22'nd'

––––––––––––––––––––––––––––

---

## Author Comment (AC1) · 4 Nov 2016

The authors thank anonymous Referee #1 for the valuable and helpful comments.

The referee suggested that "the paper could connect better with previous work on 'ripples' [Hecht] and turbulence [Yamada] and stress what's new here." As was correctly noted by the referee the new instrument features higher spatial but also temporal resolution. We emphasized this by adding the following paragraph:

"Compared to earlier airglow imagers (Hecht et al., 1997, Yamada et al., 2001) the resolution has been improved by at least one order of magnitude in both space and time. Achieving a spatial resolution of 30 m/pixel (zenith angle of 46 $^\circ$) and 17 m/pixel (zenith

angle of 0 °) the entire inertial subrange as well as the beginning viscous subrange of turbulence at airglow altitude is accessible to this instrument. Additionally, the temporal resolution of no longer than 2.8 s allows investigating the development of transient processes like breaking wave fronts." (lines 79-84)

Unfortunately, Y2001 don't give exact numbers for their horizontal resolution in units of meters, but judging from their overall field of view (FOV, 92°) and their detector size 1024 pixels, it may be estimated to be: 92°FOV/1024 pixels $\approx$ 0.09°/pixel => 87km*tan(0.09) $\approx$ 150m. H1997 smooth their initial resolution of 100km/516pixels in order to improve their signal-to-noise ratio, resulting in a resolution of approximately 750m/pixel. The temporal resolution of both Y2001 and H1997 is around one minute.

Due to these improvements the new instrument FAIM 3 can reveal features on scales significantly below one kilometre. In order to highlight these features we present two observations, the first one showing two periodic structures - and we agree with the referee, that even the larger one depicts an instability feature according to common opinion. As suggested by the referee we now refer to previous work of Hecht and Yamada on ripples and turbulence with the following new paragraph:

"While dynamical instability manifests as subordinate wave structures parallel to the initial wave fronts, convective instability emerges as wave structures perpendicular to the initial gravity wave (Andreassen et al., 1994; Fritts et al., 1994). Taylor and Hapgood, 1990 assume ripples to be the signature of KHIs, thus being related to dynamical instability. However, past observations do not agree about this issue. While Yamada et al., 2001 have presented images of instability features aligned parallel to a breaking gravity wave, Hecht et al., 1997 have observed ripple structures that were aligned perpendicular to an initial wave, which rather assigns those ripples to be caused by convective instability. In our observations none of both cases is favoured as the wave fronts of the 550 m wave packet are aligned by an angle of 45 ° to the superordinate 1.7 km wave. This supports the assumption of Hecht, 2004, that some ripples may be generated by the combination of dynamical and convective instability. According to

Fritts et al., 1996 an initially convectively driven instability structure can be rotated by the background wind shear." (lines 221-232)

In the second case study we present the turbulent breakdown of a wave front. FAIM 3 provides detailed observations of this transient event thanks to the high spatio-temporal resolution. In section 5 (discussion and summary) we added the following paragraph: "FAIM 3 not only resolves the entire inertial subrange, it also provides insight into the beginning viscous subrange of turbulence. As concerns airglow imaging this opens a new scale range of dynamic processes that can be monitored, like it is shown in the first case study. Whereas structures like the larger one (periodicity $\sim$ 1.7 km) can now be studied in greater detail with FAIM 3, structures like the smaller one (550 m) are now observable for the first time at all.

Concerning the connection of our observations with previous work in terms of scientific aspects, the second event is more evident. It shows the formation and temporal evolution of an instability feature. Due to the high temporal resolution (2.8 s) one can determine the initial formation of this structure and its later orientation relative to the initial wave field. Thus, observations of this kind are valuable for determining the nature of instability concerning the question whether such features are primarily driven convectively or dynamically.

In this context several former studies (e.g., Yamada et al., 2001, Hecht et al., 2004, Fritts et al., 1996) question whether "ripples" were initially formed parallel or perpendicular to the gravity wave fronts and then rotated by the local wind fields or formed as a combination of both instabilities. These possibilities severely complicate scientific interpretation of ripple occurrence. With the new observation capabilities provided by the FAIM 3 we can now study this initial formation in greater detail. The two instability events presented in this paper appear to be driven dynamically but in both cases there are also indications for the presence of convective instability, which suggests that these two instability mechanisms could actually accompany each other." (lines 290-305)

Of course, we are well aware of the fact, that complementary observations of the local wind field, temperature gradients etc. as provided by radars and/ or lidars are needed for a more sophisticated investigation of these cases. But the intention of our study is to give a proof of concept concerning the capabilities of highly resolved airglow observations. Furthermore the referee commented that discussing the observations with numerical modelling, "more specific discussions are needed". Since the overall aim here is to show the ability of our new imager to observe small-scale wave structures and turbulence features on scales significantly below 1 km, we decided to just give a rough qualitative comparison to numerical modelling. Closer investigations concerning the numerical simulations are beyond the scope of this paper and will be treated in future studies. In lines 308 ff. we added the paragraph

"The typical vortex structures and the decay into eddies also appear in the respective airglow modelling. Like outlined there and in the companion experimental paper (Hecht et al., 2014), the simulations have predicted such small features that could not have been resolved by airglow imagers at that time." Referring to the minor comments:

1. We added "(shortwave infrared, commonly defined from 0.9 to 1.7 $\mu$m)" behind "SWIR" in line 29 f. 2. We dropped 'frequently' in line 39. 3. Changed as suggested. New Version: "Energy and momentum transported . . . are . . ." in line 45. 4. Line 57-59: Removed as suggested. 5. We indeed mean 299 km2 as this is the horizontal area of the trapezium-shaped FOV of FAIM 3 under a zenith angle of 46 $^\circ$. The parallel sides of the trapezium are 15.2 km and 16.9 km and the height of the trapezium is 18.6 km. (line 98) 6. Yes, the temporal evolution of the image series suggests that the wave packet is advected by the background wind. The respective sentence in the paper has been extended by "[the wave packet is] apparently advected by the background wind" (line 146 f.)

7. The order N of the MEM, meaning the length of the underlying AR model when using Burg's algorithm, has been raised to a value of 36. As it has been outlined in section 3 and is described more detailed by (Wüst & Bittner, 2006), using the MEM, there is

a certain range of orders rather than the one correct order to best resolve a certain wavelength. The fact that the 550m peak stays stable after raising the order from 20 to 36 (still talking about the same data series with a length of 59) indicates that this wavelength has been fully resolved. We added "[The order] of the MEM" in line 166 f. to make it clearer.

8. Figure 6: The Referee remarked that the vortex and filaments mentioned in the text would be hard to recognize in Figure 6 and that some help would be needed. We expanded the number of images from six to fifteen. In order to better guide the reader's eye we outlined six dominating structures of this transient event and focused on describing their respective temporal evolution.

We have revised the description of the event according to the more detailed visualization in Figure 6 as follows:

"A wave front, indicated by the dashed black line in the images of Figure 6, enters the FOV in the upper right corner. While it continues propagating to the lower left, a filament separates from it on the left side (Figure 6a – b, orange). This filament moves much slower than the wave front. At around 03:18:30 UTC (Figure 6c) a second filament structure becomes visible below the first filament. In the further course of the image series (Figure 6e – f) it turns out to separate into two structures, a filament moving downward (yellow) and a stationary filament (green). At about 03:19:26 UTC (Figure 6g) the orange and the green structure begin to dissolve. The yellow structure continues propagating for a few more seconds and finally also starts decomposing at 03:19:54 UTC (Figure 6i). At 03:18:44 UTC (Figure 6d) two more filaments form at the upper right of the FOV right behind the initial wave front and are, in contrast to the other filaments, aligned perpendicular to it. They decompose at 03:19:26 UTC (Figure 6g). While the dynamics of the filaments take their course and form a vortex, rotating around a horizontally oriented axis, the initial wave front (black) overtakes the other structures, retaining its original direction (indicated by the red arrow in Figure 6h). At about 03:19:54 UTC (Figure 6i) another filament (blue) separates from it. This new

filament remains stationary and starts decaying at 03:20:50 UTC (Figure 6m). The wave front (black) keeps on propagating and leaves the FOV toward the lower left." (lines 173-185) and the caption of Figure 6 similarly.

9. Figure 7: We thank the referee for finding this mistake: the marked rectangle, which is meant to indicate the FOV of FAIM 3, is, with respect to the FAIM 4 all sky image, oriented in a wrong direction. Both cameras are adjusted in zenith position. The upper side of the FAIM 3 images corresponds to a direction of 303 ° and the upper side of the raw FAIM 4 images to a direction of 269 °. Additionally Figure 7 and Video 3 have been rotated northward and the red arrow, which indicates the direction of the propagating wave front in Figure 6, has been adopted to Figure 7. A north pointer has also been added to Figure 7, the western direction is also indicated. Instead of just marking the FAIM 3 FOV in the middle of the FAIM 4 image, we placed the respective FAIM 3 image in the revised Figure 7. Following the suggestion of Referee #2 we now also present a zoomed image (magnification factor 4).

We revised the text as follows:

"To put the observations into a larger spatial context, the FAIM 3 data are compared to simultaneous all sky measurements taken by the FAIM 4 instrument. Since the two cameras are deployed next to each other, the FOV of FAIM 3 is embedded in the centre of the FOV of FAIM 4. The FAIM 4 measurements are presented in Video 3. Besides the normal image the difference image (time difference of 60 s) is displayed on the right side; both images are rotated northward. The approximate FOV of FAIM 3 is indicated by the white boxes in Video 3. The all sky images reveal a clear and starry sky with high gravity wave activity, which can be determined on the basis of the characteristic patchy structures. The remarkable structure observed by FAIM 3 can be found again in Video 3 as a bright feature within the white box, propagating to eastern direction, which agrees with the FAIM 3 observations. Figure 7 shows the FAIM 4 all sky image at 03:20:20 UTC with the respective FAIM 3 image embedded into it (image a) as well as the image centre magnified by a factor of four (image b)." (lines 193-203) "The

aforementioned wave front is also visible in the all sky images, but a close inspection of Video 3 hardly allows perceiving indications for the separation of parts from the bright crest. Zooming into the all sky image (Figure 7b) shows that only the high-resolution measurements of FAIM 3 can reveal closer details of this structure." (lines 259-263)

"FAIM 4 all sky image taken on 5th April 2016 at 03:20:20 UTC (image a) and the magnified (zoom factor 4) image centre (image b). The entire sequence is shown in Video 3. Due to their spatial structure and their wavelength, we interpret the patchy structures in the starry sky as gravity wave fronts in the airglow layer. Comparison with the respective FAIM 3 image, which has been placed at its correct position in the middle of the all sky image shows how the small scale details of the wave crest can be resolved with the new instrument. The direction of propagation is indicated by the red arrow and matches with the observations of FAIM 3." (caption Figure 7)

Please also note the new version of Video 3, which has been submitted.

Please also note the supplement to this comment:
http://www.atmos-meas-tech-discuss.net/amt-2016-292/amt-2016-292-AC1-supplement.zip
* * *
[Figure]

[Figure]

**Fig. 1.** Figure 6 revised

2016-04-05 03:20:20.00 UTC

N

W

a

b

**Fig. 2.** Figure 7 revised

---

## Author Comment (AC2) · 4 Nov 2016

The authors thank anonymous Referee #2 for his detailed comment and the many suggestions for improvement of this paper.

The referee asked for "more discussion on what knowledge gap in these processes can be exploited or investigated with this new instrument development". He suggested elaborating the new aspects of this imager and how the observations could be used to quantitatively parameterize small scale wave instability and turbulence processes and how these could be used to improve or confirm simulations.

As the referee has stated correctly the high spatio-temporal resolution is the crucial

improvement of this airglow imager. We have elaborated the new possibilities our instrument provides more thoroughly in the following revised paragraphs:

"Compared to earlier airglow imagers (Hecht et al., 1997, Yamada et al., 2001) the resolution has been improved by at least one order of magnitude in both space and time. Achieving a spatial resolution of 30 m/pixel (zenith angle of 46 °) and 17 m/pixel (zenith angle of 0 °) the entire inertial subrange as well as the beginning viscous subrange of turbulence at airglow altitude is accessible to this instrument. Additionally, the temporal resolution of no longer than 2.8 s allows investigating the development of transient processes like breaking wave fronts." (lines 79-84)

"FAIM 3 not only resolves the entire inertial subrange, it also provides insight into the beginning viscous subrange of turbulence. As concerns airglow imaging this opens a new scale range of dynamic processes that can be monitored, like it is shown in the first case study. Whereas structures like the larger one (periodicity  $\sim$  1.7 km) can now be studied in greater detail with FAIM 3, structures like the smaller one (550 m) are now observable for the first time at all.

Concerning the connection of our observations with previous work in terms of scientific aspects, the second event is more evident. It shows the formation and temporal evolution of an instability feature. Due to the high temporal resolution (2.8 s) one can determine the initial formation of this structure and its later orientation relative to the initial wave field. Thus, observations of this kind are valuable for determining the nature of instability concerning the question whether such features are primarily driven convectively or dynamically.

In this context several former studies (e.g., Yamada et al., 2001, Hecht et al., 2004, Fritts et al., 1996) question whether "ripples" were initially formed parallel or perpendicular to the gravity wave fronts and then rotated by the local wind fields or formed as a combination of both instabilities. These possibilities severely complicate scientific interpretation of ripple occurrence. With the new observation capabilities provided by
the FAIM 3 we can now study this initial formation in greater detail. The two instability events presented in this paper appear to be driven dynamically but in both cases there are also indications for the presence of convective instability, which suggests that these two instability mechanisms could actually accompany each other." (lines 290-305)

As concerns the improvement/confirmation of simulations we inserted the paragraph "The typical vortex structures and the decay into eddies also appear in the respective airglow modelling. Like outlined there and in the companion experimental paper (Hecht et al., 2014), the simulations have predicted such small features that could not have been resolved by airglow imagers at that time." (lines 308-310)

The referee remarked that "there is a considerable amount of 'tentative' statements [...] with the interpretation of both events, particularly in the discussion and conclusions. At the end the reader is left with a couple of possibly interesting image sequences of wave events but not really sure what to make of them and what new information they can reveal".

As the referee has stated correctly it is the overall aim of this paper to demonstrate the ability of FAIM 3 to image small scale waves and instability features on scales below 1 km. When it comes to the interpretation of the observations we deliberately chose a tentative style of language because our discussion is based on bare 2D image data of airglow signals integrated from 0.9-1.7  $\mu$ m. In this paper we focus on the performance of our instrument and interpret our observations just as far as it is possible in good conscience without supporting data like background wind or temperature. However we hope after revising the discussion and conclusion section as mentioned above the significance and the outcome of the presented case studies becomes more evident to the reader.

Referring to the specific comments:

Line 30: changed to '[the hydroxyl] is the brightest component of the airglow phenomenon' AMTD
Line 136-137: Unfortunately the wave packet is quite faintly visible in the individual images. It is best apparent in the video. It is still better visible in the difference images than in the originals. We have attached Figure 2 as it would look using the original images as a supplement to this comment. In our opinion difference images provide the best help for the reader to follow the propagation of the small wave packet. We dropped 'obviously' in line 141, as suggested.

Line 154: The level of significance is calculated by performing a simple Monte Carlo test. As the referee states correctly the 550 m wave is only evident in the first half of the data series in Figure 3. At a range of 1.5 km noise predominates the wave packet. The signature of the perfect 550 m wave (orange) has been added to guide the eye without being fitted to the data.

Line 155: We agree with the referee that the peaks below 200 m might be an artefact of the sampling rate and its harmonics coupled with detector noise. Since our focus in this paper lies on analyzing evident features in the images, we decided not to further discuss this here.

Line 168-183: First of all we thank Referee #2 (as well as Referee #1 who also noticed it) for indicating that the propagation direction of the wave front in FAIM 3 (figure 6) and FAIM 4 (figure 7, all sky) did not appear to match. This made us aware of the FAIM 3 FOV not being marked correctly in Figure 7 and Video 3. The upper side of the FAIM 3 images corresponds to 303  $^{\circ}$  azimuthal direction and the upper image side of FAIM 4 to 269  $^{\circ}$  azimuthal direction.

We corrected the orientation of the FAIM 3 FOV with respect to the FAIM 4 image and rotated Figure 7 and Video 3 northward for better orientation. The Referee suggested to zoom in or blow up the FAIM 3 FOV in the FAIM 4 image to aid the comparison between the two instruments. We appreciate this idea and recognized that not only marking the FAIM 3 FOV but embedding the respective FAIM 3 image directly into the FAIM 4 image would provide the best comparison between the two imagers (see

**AMTD**
revised Figure 7). Instead of featuring the respective difference image we present an enlarged view (fourfold magnification) which shows how the FAIM 3 image fits into the FAIM 4 FOV and to what extend this new instrument is able to resolve details that could not be recognized in all sky images. As Referee #2 suggested, we also added a red arrow to Figure 7, indicating the direction of propagation of the wave front.

We revised the paper at the following instances:

"To put the observations into a larger spatial context, the FAIM 3 data are compared to simultaneous all sky measurements taken by the FAIM 4 instrument. Since the two cameras are deployed next to each other, the FOV of FAIM 3 is embedded in the centre of the FOV of FAIM 4. The FAIM 4 measurements are presented in Video 3. Besides the normal image the difference image (time difference of 60 s) is displayed on the right side; both images are rotated northward. The approximate FOV of FAIM 3 is indicated by the white boxes in Video 3. The all sky images reveal a clear and starry sky with high gravity wave activity, which can be determined on the basis of the characteristic patchy structures. The remarkable structure observed by FAIM 3 can be found again in Video 3 as a bright feature within the white box, propagating to eastern direction, which agrees with the FAIM 3 observations. Figure 7 shows the FAIM 4 all sky image at 03:20:20 UTC with the respective FAIM 3 image embedded into it (image a) as well as the image centre magnified by a factor of four (image b)." (lines 193-203)

"The aforementioned wave front is also visible in the all sky images, but a close inspection of Video 3 hardly allows perceiving indications for the separation of parts from the bright crest. Zooming into the all sky image (Figure 7b) shows that only the highresolution measurements of FAIM 3 can reveal closer details of this structure." (lines 259-263)

"FAIM 4 all sky image taken on 5th April 2016 at 03:20:20 UTC (image a) and the magnified (zoom factor 4) image centre (image b). The entire sequence is shown in Video 3. Due to their spatial structure and their wavelength, we interpret the patchy
structures in the starry sky as gravity wave fronts in the airglow layer. Comparison with the respective FAIM 3 image, which has been placed at its correct position in the middle of the all sky image shows how the small scale details of the wave crest can be resolved with the new instrument. The direction of propagation is indicated by the red arrow and matches with the observations of FAIM 3." (caption Figure 7)

Please also note the new version of Video 3, which has been submitted.

The Referee remarked that the interpretation and description of this second event was speculative and not entirely convincing to that reader and that in particular the assertion of a 'vortex' structure was very difficult to determine from the 2D image sequence.

We agree that the six images in Figure 6 are hardly sufficient to entirely display the transient event of 5th April 2016. In the revised Figure 6 we now present fifteen images of this period so that the reader can better follow the temporal evolution of the structures. We have carefully revised the description by analyzing the behaviour of six prominent features, which are also highlighted in the revised Figure 6. This should help the reader to track the filament-like structures and to follow the formation of the vortex also in the image series. From the former version of Figure 6 we have retained the red arrow indicating the direction of propagation of the overall structure, which can now also be found in the all sky image (revised Figure 7).

We have revised the following paragraph, describing the observations of the second event more detailed:

"A wave front, indicated by the dashed black line in the images of Figure 6, enters the FOV in the upper right corner. While it continues propagating to the lower left, a filament separates from it on the left side (Figure 6a – b, orange). This filament moves much slower than the wave front. At around 03:18:30 UTC (Figure 6c) a second filament structure becomes visible below the first filament. In the further course of the image series (Figure 6e – f) it turns out to separate into two structures, a filament moving downward (yellow) and a stationary filament (green). At about 03:19:26 UTC
(Figure 6g) the orange and the green structure begin to dissolve. The yellow structure continues propagating for a few more seconds and finally also starts decomposing at 03:19:54 UTC (Figure 6i). At 03:18:44 UTC (Figure 6d) two more filaments form at the upper right of the FOV right behind the initial wave front and are, in contrast to the other filaments, aligned perpendicular to it. They decompose at 03:19:26 UTC (Figure 6g). While the dynamics of the filaments take their course and form a vortex, rotating around a horizontally oriented axis, the initial wave front (black) overtakes the other structures, retaining its original direction (indicated by the red arrow in Figure 6h). At about 03:19:54 UTC (Figure 6i) another filament (blue) separates from it. This new filament remains stationary and starts decaying at 03:20:50 UTC (Figure 6m). The wave front (black) keeps on propagating and leaves the FOV toward the lower left." (lines 173-185) and the caption of Figure 6 similarly.

Referring to the suggested corrections: Line 31: omitted 'full' as suggested.

Line 34: The sentence has been expanded to "Once the waves reach the airglow layer, they influence the intensity of the airglow emission due to temperature and density variations."

Line 36: brackets omitted as suggested Line 46: We added Browning, 1971 as a reference for KHIs. Line 56: We dropped the paragraph. Line 71: Corrected to "with wavelengths down to…" Line 71: 'discovered' replaced by 'observed' as suggested. Line 71: The instruments mentioned here (FAIM 1 @ Sonnblick Observatory, Austria and FAIM 4 @ Oberpfaffenhofen, Germany) are the FAIM version described in (Hannawald et al., 2016). So the word 'similar' does not fit in this context, because they really are 'the same'.

Line 74: as suggested. 'air-craft based' changed to 'aircraft based' (now line 75) Line 80: omitted 'already mentioned' as suggested. Line 84: 'drawing' replaced by 'diagram' as suggested. Line 89: Changed to "The geometry of this arrangement implies a trapezium-shaped FOV at the airglow layer with a height of 18.6 km and a width of 15.2
km to 16.9 km.

Line 92: changed to "horizontally an overall area of about 299 km2…". We decided nevertheless to keep the word 'horizontally' in order to stress that Line 95: Omitted 'shown' and 'corresponding to 12 measured nights' as suggested. Line 98: as suggested. Line 105: In our case there is no difference between 'horizontal' and 'spatial' resolution. We decided to use 'spatial resolution'. Like the referee remarked, 'm/pixel' is of course the correct unit to describe the spatial resolution. We corrected it in lines 9, 67, 71, 97, 102, 109, 138, 168, 247 (twice), 248 (twice) and 281.

Line 107: omitted 'for example' as suggested. Line 114: Omitted the sentence as suggested and changed the beginning of the following sentence to "The periodic brightness variation related to a wavelike structure appears in the series of pixel intensities...".

Line 115: In this first analysis the direction perpendicular to the wave front has been determined by hand. Further investigation methods based on pattern recognition are currently being developed and will be applied on FAIM data in future papers.

Line 135: 'relevant period' replaced with 'interval', as suggested. Line 164: changed to "acquired at the zenith position with a spatial resolution of..." as suggested.

Line 190: dropped "if our speculation holds." Line 209-211: sentence omitted. Line 218: 'in' omitted as suggested. Line 221: see the specific comment to lines 168-183. Line 255: corrected.

Please also note the supplement to this comment: http://www.atmos-meas-tech-discuss.net/amt-2016-292/amt-2016-292-AC2supplement.zip

**AMTD**

---

## Author Comment (AC3) · 9 Nov 2016

[revised manuscript text omitted]
–of) have been acquired at the zenith position, with ensuring a spatial resolution of about 17 m/pixel. The dimensions of each image are approximately 11.1 km x 9.0 km. The zenith images show the sky as it is seen from the ground looking upwards. The upper image side corresponds to an azimuthal direction of 303°. Video 2 shows the entire event.

A wave front, indicated by the dashed black line in the images of Figure 6, enters the FOV in the upper right corner. While it continues propagating to the lower left, a filament separates from it on the left side (Figure 6a–b, orange). This filament moves much slower than the wave front. At around 03:18:30 UTC (Figure 6c) a second filament structure becomes visible below the first filament. In the further course of the image series (Figure 6e–f) it turns out to separate into two structures, a filament moving downward (yellow) and a stationary filament (green). At about 03:19:26 UTC (Figure 6g) the orange and the green structure begin to dissolve. The yellow structure continues propagating for a few more seconds and finally also starts decomposing at 03:19:54 UTC (Figure 6i). At 03:18:44 UTC (Figure 6d) two more filaments form at the upper right of the FOV right behind the initial wave front and are, in contrast to the other filaments, aligned perpendicular to it. They decompose at 03:19:26 UTC (Figure 6g). While the dynamics of the filaments take their course and form a vortex, rotating around a horizontally oriented axis, the initial wave front (black) overtakes the other structures, retaining its original direction (indicated by the red arrow in Figure 6h). At about 03:19:54 UTC (Figure 6i) another filament (blue) separates from it. This new filament remains stationary and starts decaying at 03:20:50 UTC (Figure 6m). The wave front (black) keeps on propagating and leaves the FOV toward the lower left.

A wave front, indicated by the dashed red line to guide the eye in Figure 6a, enters the FOV in the upper right corner. The straight red arrow shows the direction of propagation. A new structure develops at 03:17:48 UTC (Figure 6b), marked by the ellipse in the upper right corner. At 03:18:27 UTC (Figure 6c) some filament-like structures of different velocities arise in the area of the dashed orange box. Video 2 shows what is hardly recognizable in the displayed snapshots of Figure 6: the filaments start forming a vortex structure (emphasized by the curved yellow arrows at 03:19:01 UTC, Figure 6d), which still

To put the observations into a larger spatial context, the FAIM 3 data are compared to simultaneous all sky measurements taken by the FAIM 4 instrument. Since the two cameras are deployed next to each other, the FOV of FAIM 3 is embedded in the centre of the FOV of FAIM 4 . The FAIM 4 measurements are presented in Video 3.  Besides the normal image , the difference image (time difference of 60 s) is displayed on the right side; both images are rotated northward . The approximate FOV of FAIM 3 is indicated by the  white boxes in Video 3 . The all sky images reveal a clear and starry sky with high gravity wave activity, which can be determined on the basis of the characteristic patchy structures. The remarkable structure observed by FAIM 3 can be found again in Video 3 as a bright feature within the  white box, propagating  to eastern direction, which agrees with the FAIM 3 observations.  Figure 7 shows the FAIM 4 all sky image at 03:20:20 UTC with the respective FAIM 3 image embedded into it (image a) as well as the image centre magnified by a factor of four (image b).

**5 Discussion**

[revised manuscript text omitted]

245 The second interesting event detected by FAIM 3 is a wave front, which partly separates into filament-like features  while propagating through the FOV. Most of the filament emerge parallel to the incident wave front and are developing different velocities , so that the impression of a horizontally rotating vortex arises.  This can likely be
250 assigned to instability driven by wind shear, i.e. KHIs. The filament structure decay into disorganized features, which resemble the turbulent collapse of the wave packet on 18th November 2015. On larger scales, ther airglow observations exhibit similar-looking breakdown events of gravity wave fronts, like in (Hecht et al., 2014). The effects of KHI dynamics in the airglow layer have been modelled using DNSs and large eddy simulations  (Fritts et al. 2014). The vortex structure as well as the filament features turns out to be a typical manifestation of turbulence due to KHIs. However, in our observations
255 also some filaments aligned perpendicular to the initial wave front have been formed (Figure 6d – g, red). These indicate the presence of convective instability. The high gravity wave activity of the overall night sky, revealed by FAIM 4 all sky measurements at the same time, certainly contributes to atmospheric instability by influencing the lapse rate. Thus it could be again the combination of dynamical and convective instability which triggers the event on 5th April 2016.

The aforementioned wave front is also visible in the all sky images, but a close inspection of Video 3 hardly allows
260 perceiving indications for the separation of parts from the bright crest . Zooming into the all sky image (Figure 7b) shows that only the high-resolution measurements of FAIM 3 can reveal closer details of this structure.

**6 Summary and Conclusions**

265   In order to observe smaller scale gravity wave events and instabilities or turbulence features in the metre regime, the established airglow imaging system FAIM (Hannawald et al., 2016) has been improved with regard to spatial resolution, using an InGaAs sensor array with the fourfold number of pixels (327680) and a 100 mm SWIR lens manufactured by Edmund Optics®. The mean spatial resolution of 200 m/pixel at a 45° zenith angle and 120 m/pixel at zenith position achieved by the established FAIM system has been increased to 30 m/pixel and 17 m/pixel respectively.

270    Measurements have been taken at Oberpfaffenhofen (48.09° N, 11.28° E), Southern Germany at a zenith angle of 46° and an azimuth angle of 214° as well as in zenith position.

Two case studies are presented. On 18th November 2015 from 22:59:25 to 23:04:33 UTC a wavelike structure with a wavelength of 1.7 km and a smaller feature with a wavelength of 550 m propagate in the same direction. Their wave fronts

275   are tilted against each other by an angle of approximately 45°. The 1.7 km wave is estimated having a lifetime of about 16 min, which leads to the presumption that it might be an instability feature of a larger gravity wave with a wavelength of several kilometres which cannot be seen in our small FOV. Following this hypothesis, the 550 m structure could be the resulting subsequent instability feature. Another possibility is that the 550 m wave packet could be the primary instability feature caused by a small 1.7 km gravity wave. Both cases agree with the theory of turbulence: when atmospheric

280   instabilities lead to the breakdown of waves, their transported energy is cascaded to smaller scale structures in the inertial subrange. These considerations combined with the ratio of the wavelengths, which fits to values of earlier theoretical studies like (Klaassen and Peltier, 1991), suggest that the event of 18th November 2015 could be triggered by KHIs.

Zenith measurements on 5th April 2016 from 03:15:58 to 03:27:46 UTC exhibit the breakdown of a wave front into a vortex structure and the subsequent decay into disorganized features, probably due to turbulence. Characteristic dynamics of

285   filament-like features indicate that instability could be generated by wind shear. The observations look similar to modellings of KHI development and the consecutive turbulence dynamics of waves in the airglow layer (Fritts et al., 2014). Comparisons with parallel measurements of FAIM 4 obtaining all sky images reveal the high gravity wave activity all over the sky at that time, which might have contributed to increased atmospheric instability.

It has been demonstrated that FAIM 3 is able to image the dynamics of gravity waves on scales significantly below 1 km. FAIM 3 not only resolves the entire inertial subrange, it also provides insight into the beginning viscous subrange of turbulence. As concerns airglow imaging this opens a new scale range of dynamic processes that can be monitored, like it is shown in the first case study. Whereas structures like the larger one (periodicity ~ 1.7 km) can now be studied in greater detail with FAIM 3, structures like the smaller one (550 m) are now observable for the first time at all.

Concerning the connection of our observations with previous work in terms of scientific aspects, the second event is more evident. It shows the formation and temporal evolution of an instability feature. Due to the high temporal resolution (2.8 s) one can determine the initial formation of this structure and its later orientation relative to the initial wave field. Thus, observations of this kind are valuable for determining the nature of instability concerning the question whether such features are primarily driven convectively or dynamically.

In this context several former studies (e.g., Yamada et al., 2001, Hecht et al., 2004, Fritts et al., 1996) question whether "ripples" were initially formed parallel or perpendicular to the gravity wave fronts and then rotated by the local wind fields or formed as a combination of both instabilities. These possibilities severely complicate scientific interpretation of ripple occurrence. With the new observation capabilities provided by the FAIM 3 we can now study this initial formation in greater detail. The two instability events presented in this paper appear to be driven dynamically but in both cases there are also indications for the presence of convective instability, which suggests that these two instability mechanisms could actually accompany each other.

This can provide experimental insight into processes like turbulent breakdown of waves or generation of smaller scale wave structures due to atmospheric instabilities. FAIM 3 observations support earlier simulations of instability dynamics as presented in (Fritts et al., 2014). The typical vortex structures and the decay into eddies also appear in the respective airglow modelling. Like outlined there and in the companion experimental paper (Hecht et al., 2014), the simulations have predicted such small features that could not have been resolved by airglow imagers at that time. Apart from that, FAIM 3 is an ideal addition to all sky imagers, since it provides the opportunity to closer investigate specific airglow structures with a much higher spatial resolution, as it is demonstrated in the second case study in this paper.

Operational zenith measurements with a mean spatial resolution of 17 m/pixel and a temporal resolution of 2.8 s are being performed automatically every night since  22nd February 2016 at Oberpfaffenhofen, Germany. The data is archived at the WDC-RSAT (World Data Centre for Remote Sensing of the Atmosphere, http://wdc.dlr.de). FAIM 3 is part of the Network for the Detection of Mesospheric Change, NDMC (http://wdc.dlr.de/ndmc).

**Acknowledgements**

Parts of this research received funding from the Bavarian State Ministry of the Environment and Consumer Protection by grant number TUS01UFS-67093.

[revised manuscript text omitted]

**Figure 6. Image series of the second case study (5th April 2016 between 03:17:09 and 03:29 UTC). A wave front, indicated by the dashed  black line, propagates into the direction the red arrow in image h is indicating. While it continues propagating to the lower left, a filament separates from it on the left side (Figure 6a – b, orange). At 03:18:30 UTC (Figure 6c) a second filament structure becomes visible below the first filament. In the further course of the image series (Figure 6e – f) it turns out to separate into two structures, one filament moving downward (yellow) and one stationary filament (green). At 03:19:26 UTC (Figure 6g) the orange and the green structure begin to dissolve. The yellow structure continues**

propagating for a few more seconds and finally also starts decomposing at 03:19:54 UTC (Figure 6i). At 03:18:44 UTC (Figure 6d) two more filaments form at the upper right of the FOV right behind the initial wave front and are, in contrast to the other filaments, aligned perpendicular to it. They decompose at 03:19:26 UTC (Figure 6g). While the dynamics of the filaments take their course and form a vortex, rotating around a horizontally oriented axis, the initial wave front (black) overtakes the other structures, retaining its original direction (indicated by the red arrow in Figure 6h). At about 03:19:54 UTC (Figure 6i) another filament (blue) separates from it. This new filament remains stationary and starts decaying at 03:20:50 UTC (Figure 6m). The wave front (black) keeps on propagating and leaves the FOV toward the lower left. ~~At 03:17:48 UTC a new structure begins to develop in the elliptically shaped area (image b). Filament-like features arise (dashed orange box in image c) and form a vortex-like structure (curved yellow arrows in image d), which continues propagating into the direction of the initial wave front. The last two images show how the vortex structure decays into disorganized features while the remaining part of the wave front still follows the direction indicated by the straight red arrows (images e and f).~~ The entire sequence is shown in Video 2.

[Figure]

Figure 7. FAIM 4 all sky image taken on 5th April 2016 at 03:20:20 UTC (image a) and the magnified (zoom factor 4) image centre (image b). The entire sequence is shown in Video 3.  Due to their spatial structure and their wavelength, we interpret the patchy structures in the starry sky as gravity wave fronts in the airglow layer. Comparison with the respective FAIM 3 image, which has been placed at its correct position in the middle of the all sky image shows how the small scale details of the wave crest can be resolved with the new instrument. The direction of propagation is indicated by the red arrow and matches with the observations of FAIM 3.